# Molecular Mechanism of Overcoming Host Resistance by the *Target of Rapamycin* Gene in *Leptographium qinlingensis*

**DOI:** 10.3390/microorganisms10030503

**Published:** 2022-02-24

**Authors:** Huanli An, Tian Gan, Ming Tang, Hui Chen

**Affiliations:** State Key Laboratory for Conservation and Utilization of Subtropical Agro-Bioresources, College of Forestry and Landscape Architecture, South China Agricultural University, Guangzhou 510642, China; zzm_ahl@stu.scau.edu.cn (H.A.); gantian@stu.scau.edu.cn (T.G.); tangming@scau.edu.cn (M.T.)

**Keywords:** *Leptographium qinlingensis*, *Target of Rapamycin* gene, carbon sources, nitrogen sources, host nutrition, terpenoids

## Abstract

*Leptographium qinlingensis* is a fungal symbiont of the Chinese white pine beetle (*Dendroctonus armandi*) and a pathogen of the Chinese white pine (*Pinus armandii*) that must overcome the terpenoid oleoresin defenses of host trees to invade and colonize. *L. qinlingensis* responds to monoterpene flow with abundant mechanisms that include the decomposing and use of these compounds as a nitrogen source. Target of Rapamycin (TOR) is an evolutionarily conserved protein kinase that plays a central role in both plants and animals through integration of nutrients, energies, hormones, growth factors and environmental inputs to control proliferation, growth and metabolism in diverse multicellular organisms. In this study, in order to explore the relationship between *TOR* gene and carbon sources, nitrogen sources, host nutrients and host volatiles (monoterpenoids) in *L. qinlingensis*, we set up eight carbon source treatments, ten nitrogen source treatments, two host nutrients and six monoterpenoids (5%, 10% and 20%) treatments, and prepared different media conditions. By measuring the biomass and growth rate of mycelium, the results revealed that, on the whole, the response of *L. qinlingensis* to nitrogen sources was better than carbon sources, and the fungus grew well in maltose (carbon source), (NH_4_)_2_C_2_O_4_ (inorganic nitrogen source), asparagine (organic nitrogen source) and *P. armandii* (host nutrient) versus other treatments. Then, by analyzing the relationship between *TOR* expression and different nutrients, the data showed that: (i) *TOR* expression exhibited negative regulation in response to carbon sources and host nutrition. (ii) The treatments of nitrogen sources and terpenoids had positively regulatory effects on *TOR* gene; moreover, the fungus was most sensitive to β-pinene and 3-carene. In conclusion, our findings reveal that *TOR* in *L. qinlingensis* plays a key role in the utilization of host volatiles as nutrient intake, overcoming the physical and chemical host resistances and successful colonization.

## 1. Introduction

Bark beetle is a common pest that affects conifers [1,2], which is often associated with specific fungi. These fungi live in the special structure (reservoir) of the bark beetle or on the body surface [3,4]. In terms of killing host trees, there is an inevitable connection between them that follows certain rules. First, it is observed that the trees killed by bark beetles are stained [5,6], and symbiotic fungi can help bark beetles overcome the resistance system of host trees [7]. The reason is that the bark beetles destroy the dredging cells, block the resin tubes of host trees, kill epithelial cells, and disrupt the nutrient metabolism and water-use efficiency of hosts, finally leading to the death of host trees [8,9,10]. At the same time, the associated fungi provide more adequate nutrition for bark beetles and change the nutrition of host trees, which is conducive to the development, settlement and excavation of bark beetles [11,12]. In addition, these symbiotic fungi are also conducive to the chemical communication of bark beetles [13,14].

Chinese white pine beetle, *Dendroctonus armandi* Tsai and Li (Coleoptera: Curculionidae: Scolytidae), is the most harmful bark beetle species in the natural forest ecosystems in Northwest China [15,16]. It has mainly invaded and endangered the healthy *Pinus armandii* Franch grown in the Qinling Mountains for more than 30 years, resulting in the rapid decline of the tree potential and resistance, consequently leading to the occurrence of other diseases and pests [17,18]. It has become a major obstacle to the sustainable development of the forest ecosystem and the construction of the ecological environment in the Qinling Mountains. Controlling the invasion harm and population reproduction of *D. armandi* in the forest ecosystem, especially the implementation of population density regulation, has become an urgent theoretical and practical problem to effectively control the occurrence and harm of *D. armandi* [16,19,20].

*Leptographium qinlingensis* strain Tang and Chen is a symbiotic fungus residing in the adults of *D. armandi* [18]. When *D. armandi* invades the healthy host trees, *L. qinlingensis* is inoculated into the sapwood tissues of the phloem and xylem of trees to jointly endanger and utilize the nutrition of the host trees [20,21]. Furthermore, three toxins (6-methoxymethyleugenin, maculosin and cerevisterol) synthesized by *L. qinlingensis* are phytotoxic to the *P. armandii* seedlings [13]. In addition, inoculation with *L. qinlingensis* increases the concentrations of monoterpenes and sesquiterpenes in the phloem and xylem of the *P. armandii* seedlings [22,23,24]. These terpenoids play an important role in the settlement of *D. armandi* on *P. armandii*.

*Dendroctonus ponderosae* carrying *Grosmannia clavigera* has caused a rapid, large-scale decline of *Pinus contorta* in western North America [23,25]. Lodgepole pine also dies when inoculated at a high density of pathogenic fungi, such as *Leptographium longiclavatum*, without the beetles [23]. *G*. *clavigera* can survive independently and is induced by exposure to the lodgepole pine phloem extract (LPPE) [26] or oleoresin terpenoids [24,27]. Therefore, *L. qinlingensis* and *G*. *clavigera* belong to relative species.

Since the discovery of the Target of Rapamycin (TOR) protein in yeast cells, it has also been found in other eukaryotes such as fungi, Drosophila, plants and mammals. The function of the TOR signaling pathway is conservative among these eukaryotes, and TOR signaling is important for organism growth and development [28]. Rapamycin (Rap) is a large intracyclic ester immunosuppressant derived from *Streptomyces hygroscopicus* [29]; it can directly act on the TOR protein and inhibit its activity, so as to reduce the pathogen’s immune response [30]. In medicine, rapamycin is often used in the clinical treatment of allograft rejection [31,32]. In an in-depth study of the role of *Saccharomyces cerevisiae*, it was found that Rap can bind to and inhibit FK506-binding protein [33,34], irreversibly inhibit the G1 phase of the cell cycle, and control cell growth [35,36]. There are two TOR protein (70% homology) encoding genes in *S*. *cerevisiae* [37]. *TOR1* or *TOR2* gene can also change in the FKBP protein function-deficient strain of *S*. *cerevisiae* with Rap resistance. Knockout of *TOR1* or *TOR2* gene and Rap treatment inhibit the growth of yeast cells [38]. Therefore, *TOR*1 and *TOR*2 are considered to be the targets of the FKBP Rap complex that can inhibit TOR activity [39].

TOR is a key regulator of eukaryotic cell growth and has a large molecular weight (about 280 kDa). It is a highly conserved Serine/Threonine (Ser/Thr) protein kinase from yeasts to mammals [40] and belongs to the phosphatidylinositol kinase-related kinases (PIKKS) family [41]. The TOR signaling pathway is involved in regulating the initiation and extension of translation, ribosome production, protein biosynthesis, amino acid transport, and the transport of a variety of metabolic enzymes in eukaryotes [42,43]. It has also been found that TOR regulates intracellular metabolism, stress response, autophagy and other signaling pathways, which can affect the growth and development of fungi [44]. TOR proteins of all eukaryotic species have similar domains, and the HEAT repeats, FAT, FRB, kinase and FATC are arranged from the N-terminal to C-terminal, respectively. The N-terminal is composed of two HEAT repeat motifs. Heat repeats can form a pair of antiparallel interactions α- Helix, and HEAT repeats are the regions where the TOR complex subunit binds [45]. The FAT domain at the center and the FATC domain at the end of the C-terminal are members of the PIKK family. The interaction of these two domains may be involved in regulating kinase activity [46]. The conserved C-terminal FATC domain, as the “rapamycin target” of the kinase, is important for its own regulation and is considered to contain a peripheral membrane anchor [47]. The FRB domain is the binding region of the FKBP Rap complex. All Rap resistance caused by TOR mutation is the destruction of FRB domain [48].

Target of rapamycin complex1 (TORC1) is a highly conserved protein kinase complex and can be used as a central controller in response to environmental cues [49]. In various input signals, amino acids are effective activators and can promote a variety of anabolic reactions [50]. The TORC1 exists universally in various eukaryotes, but there are two kinds of TORC1 and TORC2 (target of rapamycin complex2) in yeasts [51]. TORC1 is composed of TOR, Raptor and LST8, and is sensitive to rapamycin (because there is an FRB subunit in TOR, which is the binding site of rapamycin) [52,53]. Moreover, TORC1 is mainly regulated by nutrient and energy utilization and participates in the process necessary to regulate protein translation and cell growth [54]. The mutation of *LST8* can affect cell growth and development but has no lethal effect on cells. It can stabilize the structure of TOR kinase [55].

In *S*. *cerevisiae*, TORC1 promotes cell growth in response to the availabilities of nitrogen and amino acids [56]. The regulation of the chromatin state is an effective strategy to quickly and reversibly control cell growth in response to the fluctuational environmental conditions; the chromatin state globally stimulates protein expression by activating ribosome biogenesis and protein translation through the AGC family kinase Sch9 [57]. This Sch9 kinase is similar to the S6 kinase (S6K1/2) in mammals and is directly phosphorylated by the TORC1 on the vacuolar membrane [58]. On the contrary, TORC1 inhibits the degradation of large proteins through the phosphorylation of Atg13 to prevent its association with the Atg1, thereby inhibiting the induction of macrophages [59]. It is proposed to show that TORC2 may have a direct impact on nuclear and chromatin function [60,61], and it is necessary for chromatin-mediated gene silencing and sub-telomere heterochromatic domain assembly [62,63].

The purpose of this study is to explore the expression patterns of the *TOR* gene in the symbiotic fungus *L. qinlingensis* of *D. armandi*, as well as the physiological roles of the *TOR* gene in helping bark beetle overcome host resistance. This study also provides new insights into the underlying molecular mechanism of *L. qinlingensis* infection in *P. armandii*.

## 2. Materials and Methods

### 2.1. Tested Strain

The tested strain in this study is *Leptographium qinlingensis* (NCBI Taxonomy ID: 717526), which was isolated from the blue transformed woody part of *P. armandii* after being invaded by *D. armandi* and was deposited at the College of Forestry, Northwest A&F University (Yangling, Xi’an, China).

The tested strain was inoculated on solid PDA medium for induction. After dark culture for 7 d, holes were drilled along the edge of the colony with a 1 cm punch, and then purified on a new MEA (0.83% malt extract powder and 0.75% technical agar, overlaid with cellophane, and the pH was adjusted to 5.5.) medium for subsequent experiments [26].

### 2.2. Main Materials

The main reagents, primers and sources used in this study were shown in Table 1.

### 2.3. TOR Gene Cloning

The TOR (Target of Rapamycin) amino acid sequence of the relative species (*G. clavigera*: a fungal associate of *D. ponderosae*) of *L. qinlingensis* and *S. cerevisiae* in the NCBI can be downloaded (https://www.ncbi.nlm.nih.gov (accessed on 14 January 2022)). Based on these sequences, the degenerate primers were obtained by the j-CODEHOP software (http://4virology.net (accessed on 14 January 2022)). *EF1* was selected as the reference gene to normalize transcript levels of *TOR* gene in *L. qinlingensis*, as it has been proven as the most stable reference gene in previous studies. Therefore, the synthesis of gene fragment sequence primers and qRT-PCR primers were referred as to the previous report [21]. The purified mycelial cultures of *L. qinlingensis* were conducted for total RNA extraction, and then the first-strand cDNA synthesis, PCR amplification, and sequencing were essentially performed.

### 2.4. Nutritional Treatments

#### 2.4.1. Medium Types

(1)Carbon Sources

In order to explore the effects of different carbon source treatments on the physiological phenotype and *TOR* gene expression of *L. qinlingensis,* 8 media were set up as follows [24,26]: CK: 1% glucose, 0.67% amino free yeast, pH = 5.5; Processing group: replaced glucose with sucrose/maltose/fructose/lactose/amylum/sorbitol/mannitol, other conditions were the same.

(2)Nitrogen Sources

In order to investigate the effects of different nitrogen sources on the physiological phenotype and *TOR* gene expression of *L. qinlingensis,* 16 media were set up (6 inorganic nitrogen sources, 4 organic nitrogen sources and 2 host nutrients) as follows [24,26,27]: CK: 0.3% NH_4_NO_3_, 0.17% amino free yeast, 0.1% potassium hydrogen phthalate, 1% maltose, pH = 5.5; processing group: replaced NH_4_NO_3_ with (NH_4_)_2_C_2_O_4_/NH_4_Cl/(NH_4_)_2_HPO_4_/(NH_4_)_2_SO_4_/(CO(NH_2_)_2_), other conditions were the same.

Note: 4 organic nitrogen sources were treated in the same way as inorganic nitrogen, host nutrient medium (10% *P. armandii*, 10% *P. tabuliformis*), other conditions were consistent.

#### 2.4.2. Effects of Different Nutritional Treatments on *L. qinlingensis* Mycelial Biomass in Liquid Culture

First, 100 mL liquid medium was placed into a 250 mL triangular flask, two fungal cakes were inoculated in each bottle and cultured in a shaking table at 28 °C and 120 r/min for 7 d. The mycelium was filtered and collected and washed with tap water 3 times, dried it at 80 °C to constant mass, and then the dry mass (biomass) of the mycelium was weighed. Five repetitions per treatment were prepared.

#### 2.4.3. Effects of Different Nutritional Treatments on *L. qinlingensis* Mycelial Growth Rate on Solid Medium

First, 1.5% agar was added to make solid medium. Before inoculation, each medium was overlaid with cellophane. One fungal cake (1 cm in diameter) was inoculated in the center of the medium and incubated at 28 °C in the dark for 15 d, The colony diameter was measured every 5 d by the cross method, and the average growth rate of mycelium was calculated. Five repetitions per treatment were prepared.

#### 2.4.4. Effects of Different Nutritional Treatments on the *L. qinlingensis TOR* Gene Expression

In the super clean workbench, the mycelium cultured for 15 d on solid medium was gently scraped with tweezers into the RNA enzyme-free 1.5 mL centrifuge tubes and then placed at −80 °C for subsequent total RNA extraction and gene expression analysis. Five repetitions per treatment were prepared.

### 2.5. Terpenoid Treatments

#### 2.5.1. Medium Types

Different terpenoids (concentrations of 5%, 10%, and 20% for each terpenoid treatment) were prepared in different medium formulations as follows [24,26]: CK: 5% DMSO, 0.83% maltose, 0.75% agar, pH = 5.5; processing group: replace DMSO with (±)-α-pinene/(−)-β-pinene/(+)-3-carene/(+)-limonene/turpentine/mix-monoterpene, other conditions were the same.

(±)-α-pinene ((+)-α-pinene: (−)-α-pinene = 1:1), monoterpene mixture ((+)-limonene: (+)-3-carene: (±)-α-pinene: (−)-β-pinene = 5:3:1:1).

#### 2.5.2. Effects of Different Terpenoid Treatments on *L. qinlingensis* Mycelial Growth Rate

First, 200 μL of the above different terpenoids were added to the MEA medium overlain with cellophane, respectively, and spread evenly with spreader, connecting it to one fungal cake, and then cultured in the dark for 15 d, and the colony diameter was measured every 5 d to determine the average growth rate of mycelium. Five repetitions per treatment were prepared.

#### 2.5.3. Effects of Different Terpenoid Treatments on the *L. qinlingensis TOR* Gene Expression

The specific methods, operations and precautions used were the same as the Section 2.4.4 in nutritional treatments. 

### 2.6. Statistical Analysis

The mycelial biomass, mycelial growth rate, and *TOR* gene expression level were statistically analyzed in this study. Five repetitions were set for each treatment, and 3 repetitions were measured to take the average values. The relative expression of *TOR* gene after qRT-PCR was calculated by 2^−^^∆∆CT^, the 2^−^^∆∆CT^ value was (log2) transformed and subjected to one-way analysis of variance (ANOVA), and Tukey’s honest significant difference test (HSD) was also used to compare treatment differences. Excel 2019 (Microsoft Office), SPSS 23.0 (IBM SPSS Statistics, Chicago, IL, USA) and SigmaPlot 12.5 software (Systat Software Inc, San Jose, CA, USA) were used for statistical analyses and plotting, respectively.

## 3. Results

### 3.1. Sequence Similarity and Phylogenetic Analysis of TOR Gene in Fungi Species

#### 3.1.1. Sequence Similarity of *TOR* Gene

The amino acid sequence (greater than 300aa) derived from the verified *TOR* gene of *L. qinlingensis* was compared with other fungal TOR proteins in the NCBI by BLASTp search to obtain the similarity between the TOR protein sequence of *L. qinlingensis* and the homologous protein sequence from other fungi species. The results were shown in Table 2.

#### 3.1.2. Phylogenetic Analysis of *TOR* Genes from Fungi Species

As shown in Figure 1, the phylogenetic tree of the *TOR* gene with the known amino acid sequences and other fungal TOR protein sequences was established by the Maximum likelihood method (Tree model: LG + G + I, −ln L = 42,370.551, G = 2.04).

According to the analysis of amino acid sequence similarity and phylogeny, the *TOR* gene of *L. qinlingensis* has the highest similarity (>90%) with *Ophiostoma piceae* and *Sporothrix brasiliensis*. In the current study, *L. qinlingensis* has high similarity and homology (>50%) with *Phyalemoniopsis curvata*, *Fusarium beominiforme*, *Colletrichum chlorophyti* and *Grosmania clavigera*.

### 3.2. Nutritional Treatments

#### 3.2.1. Effects of Different Nutritional Treatments on *L. qinlingensis* Mycelial Biomass in Liquid Medium

The results were shown in Figure 2. In Figure 2A: fructose (a) > maltose (b) > amylum (c) > glucose (d) > mannitol (e) > lactose/sorbitol (f) > sucrose (g), there were significant differences among carbon source treatments (*p* < 0.01), and fructose supply (0.2877) was significantly greater than sucrose treatment (0.0077). Figure 2B: NH_4_Cl (a) > NH_4_NO_3_ (b) > (NH_4_)_2_HPO_4_/(NH_4_)_2_C_2_O_4_ (c) > (NH_4_)_2_SO_4_ (d) > (CO(NH_2_)_2_) (e), there were significant differences among inorganic nitrogen source treatments (*p* < 0.01), and NH_4_Cl treatment (0.3080) was significantly greater than (CO(NH_2_)_2_) treatment (0.0085). Figure 2C: asparagine (a) > peptone (b) > acid hydrolyzed casein (c) > tryptone (d) > *P. armandii*/*P. tabuliformis* (e), there were significant differences among organic nitrogen and host nutrition treatments (*p* < 0.01), and asparagine supply (0.2097) was significantly greater than *P. tabuliformis* treatment (0.0463). However, *P. armandii* addition is slightly higher than *P. tabuliformis* addition, but there is no significant difference between them.

Overall, under the liquid culture conditions, the response of *L. qinlingensis* to inorganic nitrogen was better than others, and NH_4_Cl (0.3080) > fructose (0.2877) > asparagine (0.2097) > *P. tabuliformis* (0.0463). Therefore, the mycelial biomass under inorganic nitrogen treatments was greater than carbon, organic nitrogen and host nutrition sources.

#### 3.2.2. Effects of Different Nutritional Treatments on *L. qinlingensis* Mycelial Growth Rate in Solid Medium

The results were shown in Figure 3. In Figure 3A: maltose (a) > amylum (b) > sorbitol (c) > lactose/mannitol (d) > sucrose (e) > fructose (f) > glucose (g), there were significant differences among carbon source treatments (*p* < 0.01), and maltose treatment (0.1398) was significantly greater than glucose treatment (0.0524). Figure 3B: (NH_4_)_2_C_2_O_4_ (a) > NH_4_Cl (b) > (NH_4_)_2_HPO_4_ (c) > (NH_4_)_2_SO_4_ (d) > NH_4_NO_3_ (e) > (CO(NH_2_)_2_) (e), there were significant differences among inorganic nitrogen source treatments (*p* < 0.01), and (NH_4_)_2_C_2_O_4_ supply (0.1288) was significantly greater than (CO(NH_2_)_2_) supply (0.0000). Figure 3C: asparagine (a) > peptone (b) > tryptone (c) > *P. tabuliformis*/acid hydrolyzed casein (d) > *P. armandii* (e), there were significant differences among organic nitrogen and host nutrition treatments (*p* < 0.01), and asparagine treatment (0.1370) was significantly greater than the supply of *P. armandii* (0.0888).

In general, under the solid culture conditions, the response of *L. qinlingensis* to organic nitrogen was better than others, and maltose (0.1398) > asparagine (0.1370) > (NH_4_)_2_C_2_O_4_ (0.1288) > *P. armandii* (0.0888). Thus, the mycelial growth rate under carbon source treatments was greater than inorganic nitrogen, organic nitrogen and host nutrition sources.

#### 3.2.3. Effects of Different Nutritional Treatments on the *L. qinlingensis TOR* Gene Expression

The results were shown in Figure 4. In Figure 4A: lactose (a) > sucrose (b) > sorbitol (c) > mannitol (d) > fructose/maltose/amylum (de) > glucose (e), there were significant differences among carbon source treatments (*p* < 0.01), and lactose (30.3634) was significantly greater than glucose (0.7819). Figure 4B: (NH_4_)_2_C_2_O_4_ (a) > NH_4_NO_3_/(NH_4_)_2_HPO_4_ (b) > (NH_4_)_2_SO_4_/NH_4_Cl (bc) > (CO(NH_2_)_2_) (c), there were significant differences among inorganic nitrogen source treatments (*p* < 0.01), and (NH_4_)_2_C_2_O_4_ treatment (6.8893) was significantly greater than (CO(NH_2_)_2_) treatment (0.0000). Figure 4C: *P. armandii* (a) > peptone/asparagine (b) > acid hydrolyzed casein (bc) > *P. tabuliformis*/tryptone (c), there were significant differences among organic nitrogen and host nutrition treatments (*p* < 0.01), and supply of *P. armandii* (8.4987) was significantly greater than tryptone treatment (0.4178).

Broadly, under the solid culture conditions, the expression of *TOR* gene in *L. qinlingensis* was highly responsive to the carbon source treatment, and lactose (30.3634) > (NH_4_)_2_C_2_O_4_ (6.8893) > *P. armandii* (8.4987) > tryptone (0.4178). In short, the expression level of *TOR* gene in *L. qinlingensis* was highest under carbon source treatments, higher than inorganic nitrogen, host nutrition and organic nitrogen source treatments.

### 3.3. Terpenoid Treatments

#### 3.3.1. Effects of Different Terpenoid Treatments on *L. qinlingensis* Mycelial Growth Rate in Solid Medium

As shown in Figure 5, under 5% terpenoid concentration: β-pinene (a) > α-pinene/3-carene/limonene (b) > turpentine (c) > DMSO (d) > mix-monoterpene (e), there were significant differences among 5% concentration treatments (*p* < 0.01), and β-pinene (0.0950) was significantly greater than mix-monoterpene (0.0614). Under 10% terpenoid concentration: β-pinene (a) > α-pinene/3-carene (b) > limonene (c) > turpentine (d) > mix-monoterpene (e) > DMSO (f), there were significant differences among 10% concentration treatments (*p* < 0.01), and β-pinene treatment (0.1390) was significantly greater than DMSO treatment (0.1112). Under 20% terpenoid concentration: DMSO/mix-monoterpene (a) > 3-carene/limonene (b) > α-pinene (c) > β-pinene/turpentine (d), there were significant differences among 20% concentration treatments (*p* < 0.01), and DMSO treatment (0.1108) was significantly greater than turpentine treatment (0.0804).

According to data from Figure 5, we can find that, under 5% concentration treatment, the mycelial growth rate of *L. qinlingensis* was higher than the CK group (DMSO treatment), except that mix-monoterpene was lower than the CK group. Under 10% concentration treatment, all treatment groups were higher than the CK group. Under 20% concentration treatment, except that mix-monoterpene was consistent with the CK group, the other treatment groups were lower than the CK group, indicating the occurrence of the inhibitory effect. In general, under the three terpenoid concentration treatments, the mycelial growth rate of *L. qinlingensis* was weak under 5% terpenoid treatment. With an increase in terpenoid concentration, the growth rate of mycelium reached the highest under 10% terpenoid treatment and decreased significantly after 20% terpenoid treatment. However, the mycelial growth rate under 20% terpenoid treatment was still higher than 5% terpenoid supply.

#### 3.3.2. Effects of Different Terpenoid Treatments on the *L. qinlingensis TOR* Gene Expression

As shown in Figure 6, under 5% concentration: 3-carene (a) > turpentine (b) > limonene/mix-monoterpene/β-pinene (bc) > α-pinene/DMSO (c), there were significant differences among 5% concentration treatments (*p* < 0.01), and 3-carene supply (2.0788) was significantly greater than DMSO treatment (2.0788). Under 10% concentration: 3-carene (a) > β-pinene (ab) > turpentine (b) > limonene/α-pinene/mix-monoterpene/DMSO (c), there were significant differences among 10% concentration treatments (*p* < 0.01), and 3-carene supply (41.6659) was significantly greater than the DMSO group (0.8343). Under 20% concentration: 3-carene (a) > β-pinene (b) > mix-monoterpene (c) > α-pinene (cd) > turpentine (d) > DMSO (de) > limonene (e), there were significant differences among 20% concentration treatments (*p* < 0.01), and 3-carene supply (16.8437) was significantly greater than limonene treatment (0.3118).

Based on Figure 6, we can conclude that, under 5% concentration treatment, the expression of *TOR* gene in *L. qinlingensis* was higher in the treatments of 3-carene and turpentine than the CK treatment, but there was no significant difference in other treatment groups. Under 10% concentration treatment, except for 3-carene and turpentine, supply of β- pinene was also higher than the CK group. Under 20% concentration treatment, except for the results that are consistent with those of 10% concentration treatment, mix-monoterpene treatment also showed a higher effect on the *L. qinlingensis TOR* expression than the CK group.

In short, the expression of *TOR* gene in *L. qinlingensis* in the treatment group was higher than the CK group at all concentrations. The difference was that the expression of *TOR* was lower at 5% terpenoids. At the concentration of 10% terpenoids, the expression of *TOR* reached the maximum. The expression of *TOR* in the 3-carenen treatment was much higher than other treatment groups. The expression of *TOR* decreased sharply at 20% terpenoids, but the overall level of *TOR* expression was still higher than 5% terpenoids.

### 3.4. The Relationships between Different Nutrients and TOR

#### 3.4.1. Relationships between Carbon Source, Host Nutrition and *TOR*

Combined with Figure 2A, Figure 3A and Figure 4A, after treatments of fructose, maltose and amylum, the mycelial biomass and growth rate of *L. qinlingensis* were higher, but the expression of *TOR* was lower in *L. qinlingensis*, suggesting that *TOR* showed a negatively regulatory effect with the carbon source as the sole nutrition. Similarly, in Figure 2C, Figure 3C and Figure 4C, when the *P. armandii* and *P. tabuliformis* were applied as the host nutrition, the mycelial biomass and growth rate of *L. qinlingensis* were low, whereas the expression of *TOR* was high in *L. qinlingensis*. Therefore, these results revealed that *TOR* expression showed negative regulation during carbon source and host nutrition treatments.

#### 3.4.2. Relationship between Nitrogen Source and *TOR*

Integrating Figure 2B,C, Figure 3B,C and Figure 4B,C, we can conclude that either inorganic nitrogen ((NH_4_)_2_C_2_O_4_, NH_4_Cl, and NH_4_NO_3_) or organic nitrogen (asparagine and peptone) were the sole nutrient treatment, the mycelial biomass and growth rate of *L. qinlingensis* were higher, and the expression of *TOR* was also higher in *L. qinlingensis*. It is indicated that the *TOR* gene in *L. qinlingensis* may play a positive regulatory role in the nitrogen source treatments.

### 3.5. Relationship between Terpenoids and TOR

Based on Figure 5 and Figure 6, the data showed that, with the continuous increase in terpenoid concentrations (from 5% to 20%), the mycelial growth rate of *L. qinlingensis* increased, reached peak value, and then decreased, but after the treatment of 20% terpenoids, the growth rate of *L. qinlingensis* was still higher than the supply of 5% terpenoids, while the expression of *TOR* is consistent in *L. qinlingensis*. However, when terpenoids were treated as the sole nutrition, the expression of *TOR* showed a positive correlation with the increased terpenoids. It was further concluded that *TOR* could operate in *L. qinlingensis* to consume the host volatiles (such as terpenoids) as nutrition in order to overcome host resistance and to successfully colonize trees.

## 4. Discussion

In this study, the *L. qinlingensis* cultures were treated with the different nutrient elements and host-driven monoterpenoids, including carbon sources, nitrogen sources, host nutrition and terpenoids. Through comprehensive analysis by measuring the mycelial biomass, mycelial growth rate and *TOR* gene expression level in *L. qinlingensis*, we obtained that, under any nutrient treatment, the mycelium showed a positive growth effect. However, at different levels, there are differences in the mycelial biomass and growth rate of *L. qinlingensis*, which is consistent with the view of previous reports by Jüppner et al. [36,37,64], that is, *TOR* is highly conserved and responsive to nutrition in all eukaryotes, regulating cell cycle for growth. This process plays an important and ancient role in adapting to nutritional conditions, and the TOR signaling pathway is the main channel of these nutrient signals [37,65,66].

By analyzing the relationship between different nutritional treatments and *TOR* expression, we found that there was a negative regulation on *TOR* expression after carbon source and host nutritional treatments. In addition, we wonder why the mycelium of *L. qinlingensis* grew well, but the expression of *TOR* is low at the molecular level, which is in contradiction with the fact that *TOR* is the key factor that regulates carbon source and promotes cell growth and metabolism [41]. This is because carbon and nitrogen are two basic nutrient sources of all cellular organisms, which provide precursors for energy metabolism and metabolic biosynthesis [52]. In *S. cerevisiae*, different sensing and signaling pathways have been described to regulate gene expression in response to the quality of carbon and nitrogen sources [67,68]. However, in the study on the adoptive regulation of the amino acids in *S. cerevisiae*, the regulatory role of carbon catabolism repression (CCR) was clarified. When carbon and nitrogen sources exist at the same time, carbon is used as the corresponding basis for regulating the nitrogen source in order to regulate the TOR signaling pathway, and the mechanism of CCR regulating amino acid osmotic enzyme was determined [49,66,68]. In our study, only a single carbon source treatment was set up; thus, there was a “false” negative regulation of *TOR*.

For the host nutrition treatment, we used the phloem sawdust of *P. armandii* and *P. tabulaeformis* to make different medium types. The results were that the mycelial biomass and mycelial growth rate of *L. qinlingensis* showed a positive response. It can be concluded that the pathogenic fungi can successfully infect and colonize in *P. armandii* and *P. tabulaeformis*. In addition, under solid culture conditions, the mycelial growth rate of *L. qinlingensis* in *P. tabuliformis* treatment was faster than the *P. armandii* supply. As the first host of the fungus, the results were surprising. We thus speculated that there may exist novel mechanisms conducive to this situation. However, under host nutrition treatment, the expression of *TOR* was low in *L. qinlingensis*, and this was different from the characteristics of mycelial growth. Therefore, we propose that when the host phloem sawdust was treated as the sole nutrition treatment, there was no coordination of other substrates, and the key regulatory elements on the promoter of *TOR* gene failed to be activated; therefore, it showed a negative regulatory role.

TORC1 plays a central role in controlling cell growth. Nutrients activate evolutionarily conserved TORC1 through different molecular mechanisms. Nitrogen is an essential macro-nutrient element for the synthesis of amino acids, nucleotides and other cellular components [54]. In eukaryotes, there are two ways to activate the TORC1: (i) in cells, the amino acids and other nutrients stimulate their activity through Rag/Gtr GTPase, triggering the signal of Rag/Gtr dependent TORC1 activating amino acid uptake [67]; (ii) in yeast cells, TORC1 reacts to nitrogen sources through a potential mechanism and can sense and absorb several different nitrogen sources. The quality of nitrogen sources is defined by their ability to promote cell growth and glutamine accumulation, which is directly related to the ability to activate the TORC1 determined by Sch9 phosphorylation [57,58]. The preferred nitrogen source stimulates rapid and sustained Sch9 phosphorylation and glutamine accumulation. Inhibition of glutamine synthesis can reduce the activity of TORC1 and cell growth. Poor nitrogen sources stimulated rapid but transient Sch9 phosphorylation. Gtr1 deletion can prevent the transient stimulation of TORC1 but does not affect the sustained activity of TORC1. Therefore, nitrogen sources and Gtr/Rag activate TORC1 through different mechanisms [67]. In our study, during different carbon source and host nutrition treatments, mycelial growth and *TOR* expression of *L. qinlingensis* tended to be similar after nitrogen source treatment, which showed positive regulation. However, whether the TORC1 in *L. qinlingensis* follows rule (i) or (ii) as above, or whether it has its own unique regulation mechanism, needs to be further studied in future.

Insects can choose autonomously their own host, but trees as sessile organisms cannot avoid these insects. Therefore, as a long-lived static organism, conifers must resist the attacks of different and multiple attackers in their life. Consequently, conifers produce a variety of compounds resistant to diseases and pests [69]. Plant secondary chemistry is determined by both genetic and environmental factors [11]. When invaded by insects or pathogens, a large number of the secondary metabolites [9], such as terpenes, will be synthesized and released. Terpenes are organic compounds used to resist invasion and bring toxicity to insects and pathogens [23,26]. Especially in conifers, with such a large number of terpenoids and phenols, they will be resistant to various herbivores and microorganisms. Therefore, the content of terpenoids is usually used to measure the strength of plant disease resistance [15,24,70].

For this reason, in order to explore the relationship between the *TOR* gene and monoterpenoids, we designed a culture medium containing six monoterpenoids. Under the treatments of three concentrations of terpenoids [15,24], the mycelial growth of *L. qinlingensis* was positively correlated with the expression of *TOR* (see Figure 5 and Figure 6). With the continuous increase in terpenoid concentration, the mycelial growth first increased and then decreased, and the expression of *TOR* was consistent with this growth rate. The results showed that the expression of *TOR* in *L. qinlingensis* was positively correlated with the concentration of monoterpenoids. In summary, our results showed that terpenoids could induce the expression of *TOR* in the phytopathogenic fungus to promote the growth of mycelial organisms, overcoming the host physiological resistance, and then assisting the pathogens to invade and colonize successfully into trees.

## 5. Conclusions

Through different types of nutritional treatments to explore the expression patterns and roles of the *TOR* gene in *L. qinlingensis*, we obtained that the *TOR* gene is highly conserved in *L. qinlingensis*. When treated with single carbon source and host nutrition, the *TOR* gene was active and mycelia growth was also accelerated; however, under nitrogen source treatment, the mycelia grew slowly, and the expression of *TOR* was weak in *L. qinlingensis*. In order to explore the potential mechanism of the *TOR* gene in helping fungus overcome host resistance, we used host-volatile monoterpenoids in the main components to treat the *L. qinlingensis*. The results showed that the mycelial growth rate and *TOR* expression showed a synchronous trend, and that they were promoted at low concentration and inhibited at high concentration, reaching the optimum at 10% terpenoids; moreover, the fungi were highly sensitive to both β-pinene and 3-carene. In conclusion, our findings reveal that the fungal *TOR* gene can help *L. qinlingensis* to enhance its growth ability to overcome the physical and chemical host resistances and to colonize successfully into host trees.

## Figures and Tables

**Figure 1 microorganisms-10-00503-f001:**
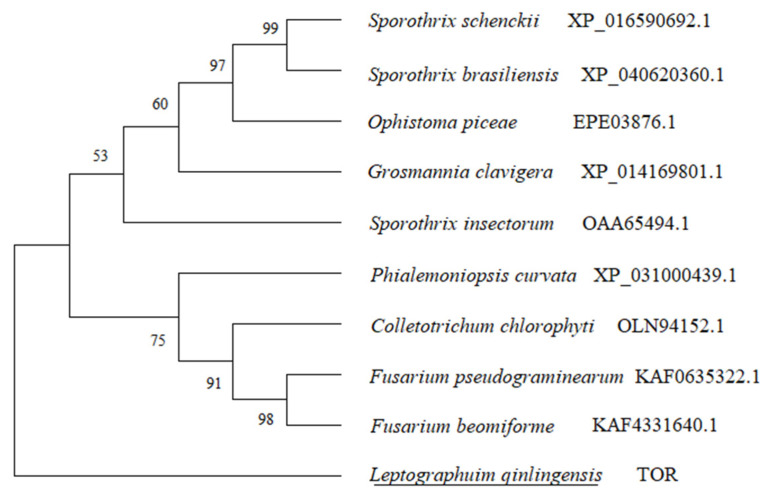
Phylogenetic analysis of the *TOR* genes from *L. qinlingensis* and other fungi with the amino acid sequences. The maximum likelihood tree was performed using the amino acidic substitution model LG + G + I (−lnL = 42,370.551, G = 2.04). The TOR amino acid sequence of *L. qinlingensis* are underlined.

**Figure 2 microorganisms-10-00503-f002:**
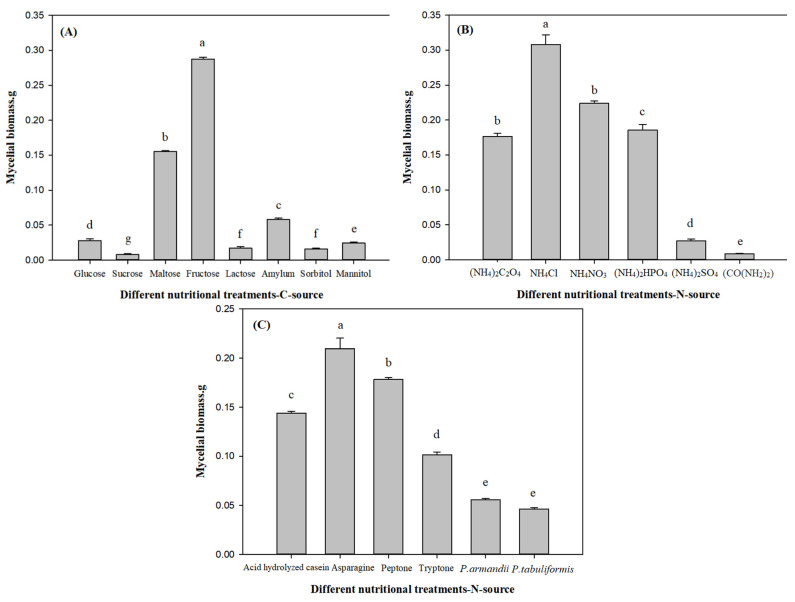
Effects of different nutritional treatments on mycelial biomass. Mycelial biomass is expressed as the mean ± S.E, and the different letters above the bars indicate significant differences (*p* < 0.01, Tukey’s HSD test). Eight biological replicates in carbon sources (**A**), 6 biological replicates in inorganic nitrogen sources (**B**), 4 biological replicates in organic nitrogen sources and 2 host nutrition biological replicates (**C**); 5 technical replicates for each biological treatment.

**Figure 3 microorganisms-10-00503-f003:**
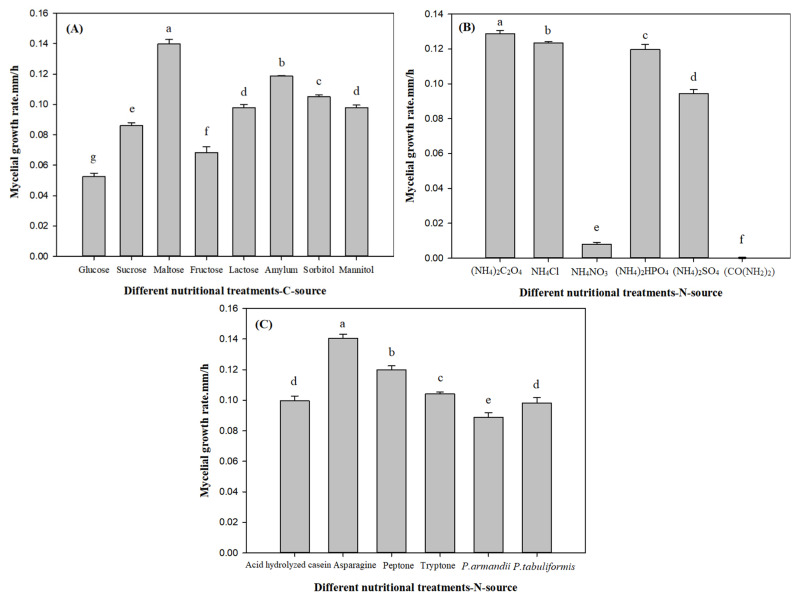
Effects of different nutritional treatments on mycelial growth rate. Mycelial growth rate is expressed as the mean ± S.E (mm/h), and the different letters above the bars indicate significant differences (*p* < 0.01, Tukey’s HSD test). Eight biological replicates in carbon sources (**A**), 6 biological replicates in inorganic nitrogen sources (**B**), 4 biological replicates in organic nitrogen sources and 2 host nutrition biological replicates (**C**); 5 technical replicates for each biological treatment.

**Figure 4 microorganisms-10-00503-f004:**
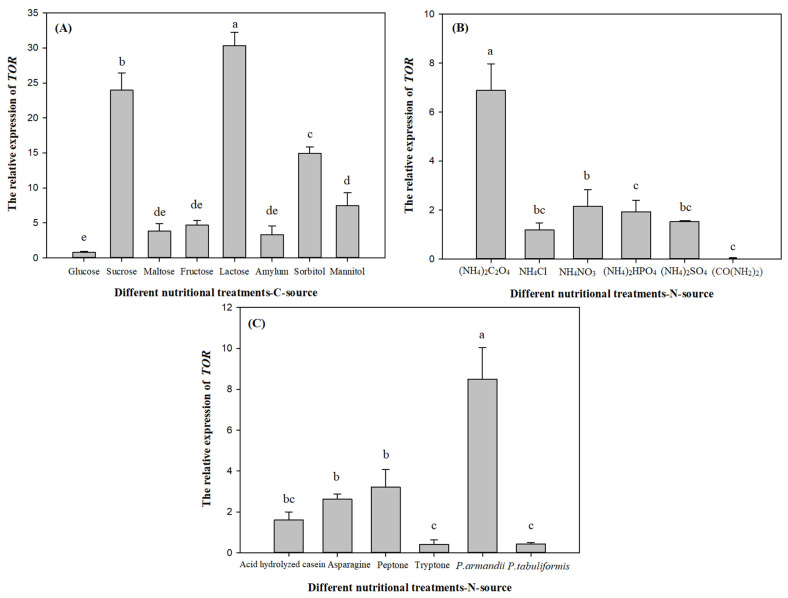
Effects of different nutritional treatments on the relative expression of *TOR*. The relative expression of *TOR* is expressed as the mean ± S.E, and the different letters above the bars indicate significant differences (*p* < 0.01, Tukey’s HSD test). Eight biological replicates in carbon sources (**A**), 6 biological replicates in inorganic nitrogen sources (**B**), 4 biological replicates in organic nitrogen sources and 2 host nutrition biological replicates (**C**); 5 technical replicates for each biological treatment.

**Figure 5 microorganisms-10-00503-f005:**
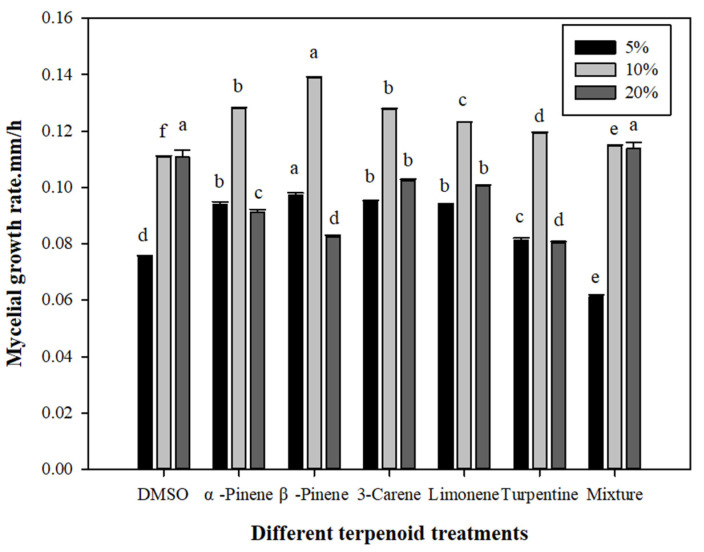
Effects of different terpenoid treatments on mycelial growth rate. Mycelial growth rate is expressed as the mean ± S.E, and the different letters above the bars indicate significant differences (*p* < 0.01, Tukey’s HSD test). Showing 5%, 10% and 20% concentration treatment, 7 biological replicates (DMSO as CK) and 5 technical replicates for each biological treatment.

**Figure 6 microorganisms-10-00503-f006:**
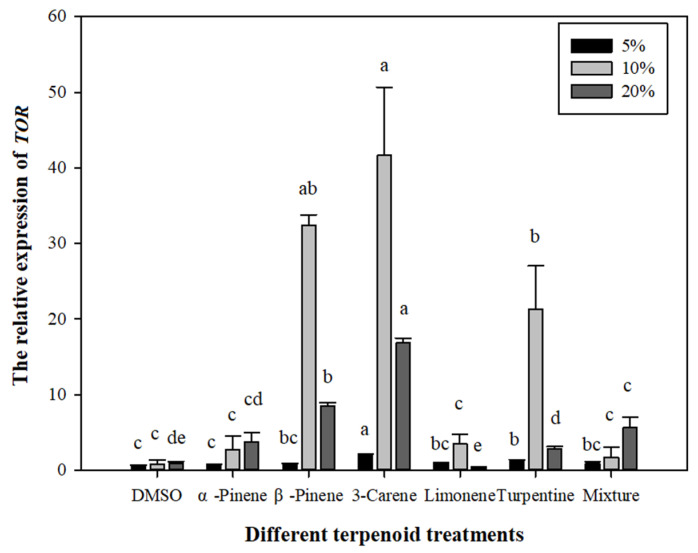
Effects of different terpenoid treatments on the relative expression of *TOR*. The relative expression of *TOR* is expressed as the mean ± S.E, and the different letters above the bars indicate significant differences (*p* < 0.01, Tukey’s HSD test). Showing 5%, 10% and 20% concentration treatment, 7 biological replicates (DMSO as CK) and 5 technical replicates for each biological treatment.

**Table 1 microorganisms-10-00503-t001:** Description of the main reagents and primers used in this study.

Key Reagents	Reagent Source
Total RNA Extractor (Trizol)	Sangon Biotech (Shanghai) Co., Ltd.
DH5 α Competent cell	Sangon Biotech (Shanghai) Co., Ltd.
HiScript^®^ III 1st Strand cDNA Synthesis Kit (+gDNA wiper)	Vazyme Biotech Co., Ltd.
HiScript^®^ III RT SuperMix for qPCR (+gDNA wiper)	Vazyme Biotech Co., Ltd.
ChamQ Universal SYBR qPCR Master Mix	Vazyme Biotech Co., Ltd.
Trelief^TM^ SoSoo Cloning Kit Ver.2	Tsingke Biotechnology Co., Ltd.
E.Z.N.A. Gel Extraction Kit	*Omega* Bio-Tek Co., Ltd.
DMSO/turpentine	Moklin Biotechnology Co., Ltd.
(+)-α-pinene/(−)-α-pinene	Shanghai Aladdin Bio-Technology Co., Ltd.
(−)-β-pinene/(+)-3-carene	Shanghai Aladdin Bio-Technology Co., Ltd.
(+)-limonene/mix-monoterpene	Shanghai Aladdin Bio-Technology Co., Ltd.
*P. armandii* Franch	Northwest A&F University (Yangling, China)
*P. tabuliformis* Carr	Northwest A&F University (Yangling, China)
Gene (Primer)	
*TOR*: Fragment amplification	
F: GGAACTTCTCCCGGGTCATG	Sangon Biotech (Shanghai) Co., Ltd.
R: GGTGGCCATCCTGTGGCACG	Sangon Biotech (Shanghai) Co., Ltd.
q-PCR F: TCTCCTTAACATTGAGCACCG	Sangon Biotech (Shanghai) Co., Ltd.
R: ATAGCCAAACACCTCCACC	Sangon Biotech (Shanghai) Co., Ltd.
*EF1*: Fragment amplification	
F: GCTGCTGTCCGTGTTGAA	Sangon Biotech (Shanghai) Co., Ltd.
R: GGTTGTAGCCGACCTTCTT	Sangon Biotech (Shanghai) Co., Ltd.
q-PCR F: CTTGGTGGTGTCCATCTTGTT	Sangon Biotech (Shanghai) Co., Ltd.
R: CCGCTGGTACGGGTGAGTT	Sangon Biotech (Shanghai) Co., Ltd.

**Table 2 microorganisms-10-00503-t002:** Amino acidic identity of the *TOR* gene from *L. qinlingensis*, with the relative sequences in other fungi.

Gene	Blastp Matches in Gene Bank		Identity%
	Species	Accession No.	
TOR2-kinase	*Ophiostoma piceae*	EPE03876.1	93.35
TOR-kinase	*Sporothrix brasiliensis*	XP_040620360.1	92.75
TOR-kinase	*Grosmannia clavigera*	XP_014169801.1	88.22
TOR-kinase	*Sporothrix insectorum*	OAA65494.1	88.22
TOR2-kinase	*Fusarium culmorum*	PTD02212.1	84.89
TOR2-kinase	*Fusarium graminearum*	PCD20916.1	84.89
TOR2-kinase	*Colletotrichum chlorophyti*	OLN94152.1	85.50
TOR2-kinase	*Fusarium oxysporum*	KAG7412775.1	84.59
TOR-kinase-like	*Trichoderma longibrachiatum*	PTB77317.1	85.50
TOR 2-kinase	*Colletotrichum tanaceti*	TKW85705.1	85.80
TOR 2-kinase	*Colletotrichum incanum*	KZL74444.1	85.50
TOR-kinase	*Trichoderma parareesei*	OTA01228.1	85.50
TOR 2-kinase	*Colletotrichum aenigma*	XP_037171942.1	85.50
TOR 2-kinase	*Colletotrichum asianum*	KAF0318508.1	85.50
TOR 2-kinase	*Colletotrichum shisoi*	TQN71010.1	85.80
TOR-kinase-like	*Trichoderma reesei*	XP_006964956.1	85.50
TOR 2-kinase	*Colletotrichum siamense*	XP_036488359.1	85.50
TOR 2-kinase	*Colletotrichum viniferum*	KAF4919045.1	85.50
TOR 2-kinase	*Colletotrichum camelliae*	KAH0430715.1	85.50
TOR 2-kinase	*Colletotrichum incanum*	OHW92381.1	85.50

## Data Availability

Not applicable.

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
