# Peer review of "Molecular Mechanism of Overcoming Host Resistance by the Target of Rapamycin Gene in Leptographium qinlingensis"

_microorganisms, 2022, doi:10.3390/microorganisms10030503_

Round 1
Reviewer 1 Report
The manuscript " Molecular Mechanism of Overcoming Host Resistance by TOR (Target of Rapamycin) Gene of Leptographium qinlingensis analyzes the role of TOR gene in the symbiotic fungus L. qinlingensis of D. armandi, and the physiological function of TOR gene in helping the bark beetle to overcome host resistance, in order to further reveal the mechanism of infection of L. qinlingensis to P. armandii.
For this aim, the relationship between the TOR gene and carbon sources, nitrogen sources, host nutrition, and host volatiles (terpenoids) in L. qinlingensis was analyzed. Different treatments were set up using various carbon sources, nitrogen sources, main host nutrition, different treatments with monoterpenoids at several concentrations (5%, 10%, and 20%), and also different types of media were prepared. The article "Molecular mechanism of overcoming host resistance by TOR (Target of Rapamycin) gene of Leptographium qinlingensis analyzes the role of TOR gene in the symbiotic fungus L. qinlingensis of D. armandi, and the physiological function of TOR gene in helping the bark beetle to overcome host resistance, in order to further reveal the mechanism of infection of L. qinlingensis to P. armandii.
For this aim, the relationship between the TOR gene and carbon sources, nitrogen sources, host nutrition, and host volatiles (terpenoids) in L. qinlingensis was analyzed. Different treatments were set up using various carbon sources, nitrogen sources, main host nutrition, different treatments with monoterpenoids at several concentrations (5%, 10%, and 20%), and also different types of media were prepared.
By measuring mycelial biomass and growth rate, it was observed that the response of L. qinlingensis to nitrogen sources was improved over carbon sources, and the fungus grown on maltose, (NH4)2C2O4, asparagine, and P. armandii grew better. The authors also observed that TOR showed negative regulation in the presence of carbon sources and host nutrition, otherwise nitrogen sources and terpenoids had a positive regulatory effect, furthermore, the fungi were the most sensitive to ß- pinene and 3-carene. The authors in this work demonstrate that, in L. qinlingensis, TOR plays a key role in utilizing host volatiles as nutrients, overcoming host physical and chemical resistance, and successful colonization.
The manuscript is certainly well-written and innovative, and the results advance current knowledge of the behavior of the fungus L. qinlingensis. The results are presented in relevant manner and are significant. All conclusions are justified and supported by the results.
The manuscript is written clearly and completely and the data and analysis are presented accurately. The study is properly designed and technically sound, the analyses are performed to high technical standards, and the data are robust enough to draw conclusions. The methods, instruments, software, and reagents are described in sufficient detail.
Minor revision:
- Please move the fragment rows 70-75 after the description of the role played by TOR. The Rapamycin (Rap) description can be put after that.
- Please increase the size of figures (2, 3, 4, 5 and 6) are small.
- Rows 309-310 move in discussion : “In addition, we were surprised that under solid culture conditions, the mycelial growth rate of qinlingensis in P. tabuliformis was faster than P. armandii. As the first host of fungi, the results were obviously surprising. Therefore, we speculated that there should be some mechanism leading to this situation".
- Please check the correct nomenclature of reference: such as row 430 “ Jessica et al …….”. Jessica is a name not surname.
- check the text formatting.
Author Response
Reviewer 1
The manuscript " Molecular Mechanism of Overcoming Host Resistance by TOR (Target of Rapamycin) Gene of Leptographium qinlingensis analyzes the role of TOR gene in the symbiotic fungus L. qinlingensis of D. armandi, and the physiological function of TOR gene in helping the bark beetle to overcome host resistance, in order to further reveal the mechanism of infection of L. qinlingensis to P. armandii.
For this aim, the relationship between the TOR gene and carbon sources, nitrogen sources, host nutrition, and host volatiles (terpenoids) in L. qinlingensis was analyzed. Different treatments were set up using various carbon sources, nitrogen sources, main host nutrition, different treatments with monoterpenoids at several concentrations (5%, 10%, and 20%), and also different types of media were prepared. The article "Molecular mechanism of overcoming host resistance by TOR (Target of Rapamycin) gene of Leptographium qinlingensis analyzes the role of TOR gene in the symbiotic fungus L. qinlingensis of D. armandi, and the physiological function of TOR gene in helping the bark beetle to overcome host resistance, in order to further reveal the mechanism of infection of L. qinlingensis to P. armandii.
For this aim, the relationship between the TOR gene and carbon sources, nitrogen sources, host nutrition, and host volatiles (terpenoids) in L. qinlingensis was analyzed. Different treatments were set up using various carbon sources, nitrogen sources, main host nutrition, different treatments with monoterpenoids at several concentrations (5%, 10%, and 20%), and also different types of media were prepared.
By measuring mycelial biomass and growth rate, it was observed that the response of L. qinlingensis to nitrogen sources was improved over carbon sources, and the fungus grown on maltose, (NH4)2C2O4, asparagine, and P. armandii grew better. The authors also observed that TOR showed negative regulation in the presence of carbon sources and host nutrition, otherwise nitrogen sources and terpenoids had a positive regulatory effect, furthermore, the fungus was the most sensitive to β-pinene and 3-carene. The authors in this work demonstrate that, in L. qinlingensis, TOR plays a key role in utilizing host volatiles as nutrients, overcoming host physical and chemical resistance, and successful colonization.
The manuscript is certainly well-written and innovative, and the results advance current knowledge of the behavior of the fungus L. qinlingensis. The results are presented in relevant manner and are significant. All conclusions are justified and supported by the results.
The manuscript is written clearly and completely and the data and analysis are presented accurately. The study is properly designed and technically sound, the analyses are performed to high technical standards, and the data are robust enough to draw conclusions. The methods, instruments, software, and reagents are described in sufficient detail.
Thank you for taking time out of your busy schedule to review the manuscript. Now we have carefully corrected and replied the manuscript for this revision. The revision instructions are as follows:
- Q-1: Please move the fragment rows 70-75 after the description of the role played by TOR. The Rapamycin (Rap) description can be put after that.
- A-1: Since the discovery of the Target of Rapamycin (TOR) protein in yeast cells, it has also been found in other eukaryotes such as fungi, Drosophila, plants and mammals. The function of TOR signaling pathway is conservative among these eukaryotes, and the TOR signaling is very important for the organisms’ growth and development [28]. Rapamycin (Rap) is a large intracyclic ester immunosuppressant derived from Streptomyces hygroscopicus [29], it can directly act on TOR protein and inhibit its activity, so as to reduce the pathogen's immune response [30]. In medicine, rapamycin is often used in the clinical treatment of allograft rejection [31,32].
- Q-2: Please increase the size of figures (2, 3, 4, 5 and 6) are small.
- A-2: The figures (2, 3, 4, 5 and 6) have been enlarged.
- Q-3: Rows 309-310 move in discussion: “In addition, we were surprised that under solid culture conditions, the mycelial growth rate of qinlingensisin P. tabuliformis was faster than P. armandii. As the first host of fungus, the results were obviously surprising. Therefore, we speculated that there should be some mechanism leading to this situation".
- A-3: In addition, it was surprised that under solid culture conditions, the mycelial growth rate of qinlingensis in P. tabuliformis treatment was faster than P. armandii supply. And, as the first host of fungus, the results were obviously surprising. We thus speculated that there may exist novel mechanisms conducting this situation. This paragraph has been revised and put into discussion.
- Q-4: Please check the correct nomenclature of reference: such as row 430 “Jessica et al …….”. Jessica is a name not surname.
- A-4: Changed to surname (Jessica has been replaced by Jüppner).
- Q-5: Check the text formatting.
- A-5: The text formatting has been checked carefully.
Thank you very much for your valuable comments and suggestions. To sum up, based on your suggestions and comments, we have revised and checked it carefully. At the same time, the language and style have also been slightly adjusted. Please do not hesitate to contact us if there are any question. Thank you again for your hard work! Best wishes to you!
Reviewer 2 Report
Manuscript microorganisms-1577757 "Molecular Mechanism of Overcoming Host Resistance by TOR (Target of Rapamycin) Gene of Leptographium qinlingensis" by An and collaborators describes the effect of nutrient sources on mycelial growth and TOR expression. This is relevant as L. qinlingensis is a pathogen of an important pine tree used for reforestation. The study presents a thorough introduction and is well described, with appropriate statistical methods. Overall it is an important contribution but before being published, it must be reviewed in detail by a native English speaker to adjust it to the quality standards of MDPI. Below I made just a couple of minor suggestions but there are many adjustments necessary throughout the text to improve clarity.
Line 23-26: This sentence is too long and not easy to follow. I recommend separating in 2.
Check every citation of L. qinlingensis, some are currently not in italic.
Author Response
Reviewer 2
Comments and Suggestions for Authors Manuscript microorganisms-1577757 "Molecular Mechanism of Overcoming Host Resistance by TOR (Target of Rapamycin) Gene of Leptographium qinlingensis" by An and collaborators describes the effect of nutrient sources on mycelial growth and TOR expression. This is relevant as L. qinlingensis is a pathogen of an important pine tree used for reforestation. The study presents a thorough introduction and is well described, with appropriate statistical methods. Overall, it is an important contribution but before being published, it must be reviewed in detail by a native English speaker to adjust it to the quality standards of MDPI. Below I made just a couple of minor suggestions but there are many adjustments necessary throughout the text to improve clarity.
Thank you for taking time out of your busy schedule to review the manuscript. Now we have carefully corrected and replied the manuscript for this revision. The revision instructions are as follows:
- Q-1: Line 23-26: This sentence is too long and not easy to follow. I recommend separating in 2.
- A-1: And then, by analyzing the relationship between TOR expression and different nutrients, the data showed that: (i) TOR expression exhibited negative regulation in response to carbon sources and host nutrition. (ii) The treatments of nitrogen sources and terpenoids had positively regulatory effects on TOR gene, moreover, the fungus was most sensitive to β-pinene and 3-carene. This sentence has been revised carefully.
- Q-2: Check every citation of qinlingensis, some are currently not in italic.
- A-2: The text formatting has been checked carefully.
Thank you very much for your valuable comments and suggestions. To sum up, based on your suggestions and comments, we have revised and checked it carefully. At the same time, the language and style have also been slightly adjusted. Please do not hesitate to contact us if there are any question. Thank you again for your hard work! Best wishes to you!